# Accuracy comparison of ARIMA and XGBoost forecasting models in predicting the incidence of COVID-19 in Bangladesh

**Md. Siddikur Rahman** * , **Arman Hossain Chowdhury** , **Miftahuzzannat Amrin**

Department of Statistics, Begum Rokeya University, Rangpur, Bangladesh

☯ These authors contributed equally to this work.

* siddikur@brur.ac.bd

**Data Availability Statement:** All data are in the manuscript and/or supporting information files.

**Funding:** The authors received no specific funding for this work.

## Abstract

Accurate predictive time series modelling is important in public health planning and response during the emergence of a novel pandemic. Therefore, the aims of the study are three-fold: (a) to model the overall trend of COVID-19 confirmed cases and deaths in Bangladesh; (b) to generate a short-term forecast of 8 weeks of COVID-19 cases and deaths; (c) to compare the predictive accuracy of the Autoregressive Integrated Moving Average (ARIMA) and eXtreme Gradient Boosting (XGBoost) for precise modelling of non-linear features and seasonal trends of the time series. The data were collected from the onset of the epidemic in Bangladesh from the Directorate General of Health Service (DGHS) and Institute of Epidemiology, Disease Control and Research (IEDCR). The daily confirmed cases and deaths of COVID-19 of 633 days in Bangladesh were divided into several training and test sets. The ARIMA and XGBoost models were established using those training data, and the test sets were used to evaluate each model's ability to forecast and finally averaged all the predictive performances to choose the best model. The predictive accuracy of the models was assessed using the mean absolute error (MAE), mean percentage error (MPE), root mean square error (RMSE) and mean absolute percentage error (MAPE). The findings reveal the existence of a nonlinear trend and weekly seasonality in the dataset. The average error measures of the ARIMA model for both COVID-19 confirmed cases and deaths were lower than XGBoost model. Hence, in our study, the ARIMA model performed better than the XGBoost model in predicting COVID-19 confirmed cases and deaths in Bangladesh. The suggested prediction model might play a critical role in estimating the spread of a novel pandemic in Bangladesh and similar countries.

## Introduction

The coronavirus disease 2019 (COVID-19) is a major global public health threat. A group of pneumonia infections caused by a newly found β-coronavirus occurred in Wuhan, China in December 2019 [1]. On January 12, 2020, the World Health Organization (WHO) labelled this coronavirus the 2019-novel coronavirus (2019-nCoV) [2, 3]. More than 222 nations, including Bangladesh, have reported more than 263.1 million confirmed COVID-19 cases as of

**Competing interests:** The authors have declared that no competing interests exist.

November 30, 2021, resulting in 5.2 million fatalities worldwide [4]. On March 8, 2020, IEDCR detected the first COVID-19 case in Bangladesh. On March 9, 2020, the number of infected cases began to rise, and as of December 31, 2021, Bangladesh had 1.6 million infected cases and 28,072 fatalities [5].

In South Asia, especially Bangladesh, COVID-19 has portrayed a significant gap in public health preparedness and response to contagious disease risks and outbreaks [6]. The lack of a dependable public health surveillance system is noticeable [7]. Of the 222 countries, Bangladesh globally ranks 4th on the daily increase of COVID-19 deaths [8] and 3rd in fatality rate in South Asia [9]. Bangladesh is a densely populated country, with almost 161.4 million people living in overcrowded cities and villages, with a population density of over 1115 persons per square kilometre [10]. The healthcare system of Bangladesh is falling short of international standards due to a scarcity of competent workers and inadequate healthcare services, despite the Bangladesh government's efforts to address these challenges in the health service [11]. Furthermore, there are insufficient Intensive Care Unit (ICU) beds for the population. The government faces an uphill battle to control the COVID-19 spread. The Impact of COVID-19 in Bangladesh on education is also noticeable. Due to the lengthy university shutdown and home confinement caused by COVID-19, students' learning was severely disrupted [12]. Students had a higher psychological effect due to COVID-19 [13]. The spread of COVID-19 poses a tremendous challenge for any administration in terms of public health system capacity and management in the event of a catastrophic emergency [14]. As a result, knowing the exact prediction and usual pattern of this virus is crucial for Bangladesh. The prediction model can assist hospitals, healthcare administration and related stakeholders in public health planning and response during the emergence of the COVID-19 pandemic.

The autoregressive integrated moving average (ARIMA) model is commonly used in the modelling of contagious diseases [15], such as influenza viruses [16], malaria [17], and hemorrhagic fever [18]. Several studies regarding COVID-19 forecasting used ARIMA model for predicting the confirmed cases and examined it as the best model [19–22]. On the other hand, the eXtreme Gradient Boosting, a new approach, is an uptrend machine learning technique in time series modelling [23, 24]. The XGBoost model has performed admirably in many medical research sectors [25–28], but the application of XGBoost model in predicting COVID-19 incidence is scanty [29–32]. Time series forecasting methods play a critical role in estimating the spread of an epidemic. Therefore, this study aimed to (a) model the overall trend of COVID-19 confirmed cases and deaths in Bangladesh; (b) generate a short-term forecast of 8 weeks of confirmed COVID-19 cases and deaths; (c) compare the predictive accuracy of the Autoregressive Integrated Moving Average (ARIMA) and eXtreme Gradient Boosting (XGBoost) for precise modelling of non-linear features and seasonal trends of the time series (Fig 1). The findings of this study will help policymakers and government officials with effective public health interventions to control the spread of an epidemic.

## Methods

### Data source

Daily confirmed cases and deaths of COVID-19 in Bangladesh from March 08, 2020, to November 30, 2021 were collected from the Directorate General of Health Service (DGHS) and Institute of Epidemiology, Disease Control and Research (IEDCR) [33, 34]. The daily confirmed cases and deaths of COVID-19 of 633 days in Bangladesh were divided into several training and test sets. The ARIMA and XGBoost models were established using those training data, and the test sets were used to evaluate each model's ability to forecast and finally averaged all the predictive performances to choose the best model.

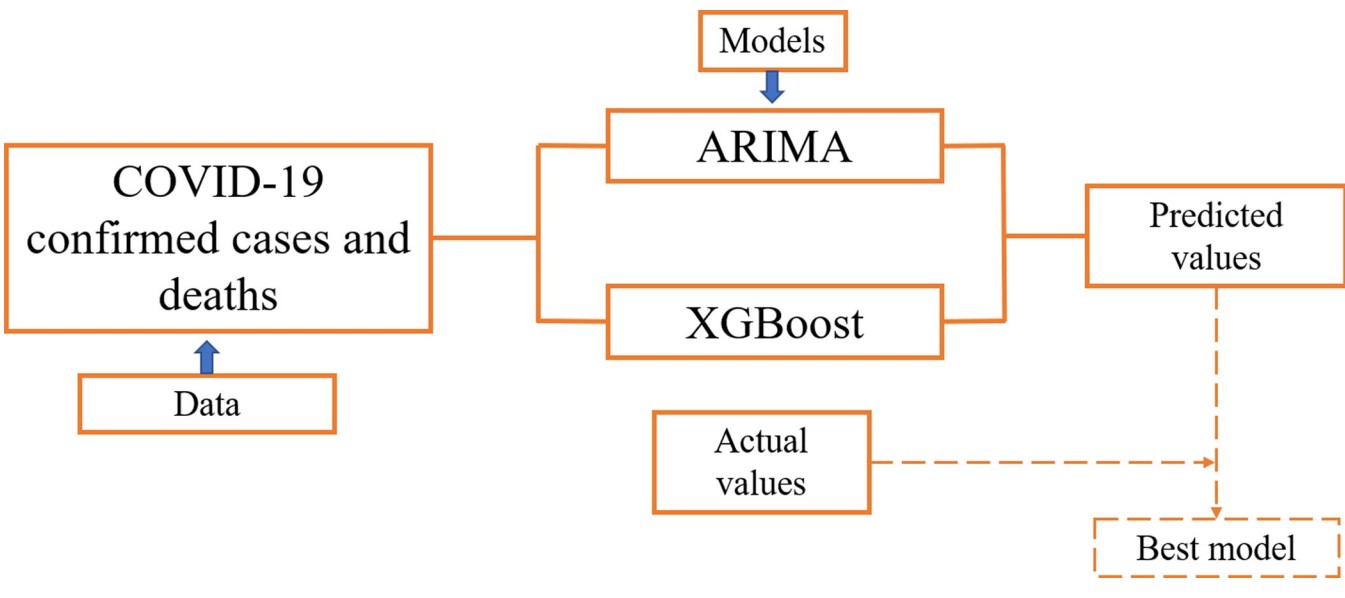

**Fig 1. Proposed methodology.**

## ARIMA model

The ARIMA model is frequently used for time series modelling of contagious diseases [35]. It is one of the most often used time-series models in a variety of sectors of data analysis because it accounts for changing trends, periodic variations, and random disturbances in the data. It's utilized for forecasting and better interpreting the data [36]. ARIMA(p, d, q) is a combination of the Autoregressive (AR) and Moving Average (MA) models, with the 'I' standing for integration; where p denotes the autoregressive order, d for differencing order, and q for moving average order [37]. Stationary is a discardable property for a time series analysis. The difference order d is used to make a nonstationary time series to stationary. It is estimated by the Augmented Dickey-Fuller (ADF) test. An ARMA (p, q) model combines AR(p) and MA(q) models, which is best suited to univariate time series analysis. The AR(p) model assumes that a variable's future value is determined by a linear combination of p previous observations plus a random error term. The AR(p) model is represented mathematically as follows:

$$Y_t = C + \emptyset_1 Y_{t-1} + \emptyset_2 Y_{t-2} + \emptyset_3 Y_{t-3} + \emptyset_4 Y_{t-4} \ldots \ldots \emptyset_p Y_{t-p} + \varepsilon_t \tag{1}$$

$Y_t$ and $\varepsilon_t$ denote the actual value and error terms at time t, $\emptyset_i$ (i = 1,2,3,4. . ..) denotes model parameters, and c denotes a constant. The order of the model is a positive integer p. Unlike the AR(p) model, the MA(q) model includes a dependent variable for previous errors. Following is the MA(q) model:

$$Y_t = \mu + \theta_1 \varepsilon_{t-1} + \theta_2 \varepsilon_{t-2} + \theta_3 \varepsilon_{t-3} + \theta_4 \varepsilon_{t-4} + \cdots + \theta_q \varepsilon_{t-q} + \varepsilon_t \tag{2}$$

Here, $\mu$ denotes the series' mean, $\theta_j$ (j = 1, 2, 3. . . q) denotes model parameters, and q is the model's order. A mathematical representation of an ARMA (p, q) model is as follows:

$$Y_t = C + \mu + \emptyset_1 Y_{t-1} + \emptyset_2 Y_{t-2} + \emptyset_3 Y_{t-3} + \emptyset_4 Y_{t-4} \ldots \ldots + \emptyset_p Y_{t-p} + \theta_1 \varepsilon_{t-1} + \theta_2 \varepsilon_{t-2} + \theta_3 \varepsilon_{t-3}$$
$$+ \theta_4 \varepsilon_{t-4} + \cdots + \theta_q \varepsilon_{t-q} + \varepsilon_t \tag{3}$$

## Seasonal ARIMA model

A seasonal ARIMA model collects information from seasonal components that the conventional ARIMA model cannot comprehend. The seasonal model may be split into two types based on its complexity: an additive model (simple seasonal model) and a product seasonal model. The mathematical expression of the simple seasonal model's is:

$$X_t = S_t + T_t + I_t \qquad (4)$$

Where, $S_t$, $T_t$, and $I_t$ denote seasonal information, trend information, and random fluctuation information in the data, respectively. To build a seasonal ARIMA model, the components of the nonseasonal part is identified first. After that, the seasonal part is identified. For the seasonal information, the time series data were plotted to see the seasonality pattern. Then the Box-Cox transformation was performed to reduce the variance of the original COVID-19 time series. At the same time, the long term trend and seasonal variations were fixed by performing first-order differencing and seasonal differencing. An Augmented Dickey-Fuller (ADF) test can be used to determine if the time series is stable. The potential values of the autoregressive order p, moving average order q, seasonal autoregressive order P, and the seasonal moving average order Q may be calculated using the graphs of the autocorrelation function (ACF) and partial ACF (PACF) determined by the Box- Jenkins order determination method [38]. The corrected Akaike information criterion (AICc) value was used to evaluate the benefits and drawbacks of the model fit, and the model with its least AICc value was deemed the best. The Ljung-Box test is thus used to determine the white noise of the residuals [18, 38].

## XGBoost model

Extreme Gradient boosting (XGBoost) technique is an optimized distributed Gradient boosting library that can rapidly assess the importance of all input features and is a scalable machine learning system for tree boosting. It has proven to be a qualified and competent problem solver for machine learning [39, 40]. Gradient boosting is a popular method for building a forecasting model and a quantifiable boosting algorithm [38]. It was initially developed by Chen Tianqi and Carlos Gestrin in 2011 and has since been improved and polished by numerous scientists in the follow-up study [41]. The core concept of boosting (enhancing machine learning models) is to merge hundreds of low-accuracy prediction models into a single high-accuracy model. Several models must frequently be integrated to obtain good prediction accuracy under tolerable parameter values. The model may need to be iterated or repeated multiple times or more to attain sufficient accuracy if the data collection is vast or complicated; the XGBoost model could better handle this problem [18]. XGBoost is a robust and effective gradient boosting machine algorithm [42, 43]. The objective function can be written as follows:

$$Obj^{(t)} = \sum_{i=1}^{n} l(y_i, \hat{y}_i^{(t-1)} + f_t(x_i)) + \Omega(f_t) + constant \qquad (5)$$

Where $y_i$ is the observed values, $\hat{y}_i^{(t-1)}$ is the predicted value of the last iteration, $x_i$ is the feature vector, n is the sample size, $f_t$ is a new function which model learns, $\Omega(f_t)$ is the regularization term which saves the model from complexity. $l$ denotes the loss function, which calculates the difference between the label and the prediction in the previous phase, the new tree's output [38, 44].

## Evaluation parameter of models

A model's real accuracy can be measured by comparing predicted and actual values. A variety of performance metrics can be performed to calculate accuracy [45]. We used four prominent

forecasting parameters to assess the predictive efficacy of our model: Mean Absolute Error (MAE), Root Mean Square Error (RMSE), and Mean Absolute Percentage Error (MAPE), Mean Percentage Error (MPE), as follows:

$$MAE = \frac{1}{n}\sum_{i=1}^{n}|\hat{y}_i - y_i| \tag{6}$$

$$RMSE = \sqrt{\frac{1}{n}\sum_{i=1}^{n}(\hat{y}_i - y_i)^2} \tag{7}$$

$$MAPE = \frac{1}{n}\sum_{i=1}^{n}|\frac{\hat{y}_i - y_i}{y_i}| \times 100\% \tag{8}$$

$$MPE = \frac{1}{n}\sum_{i=1}^{n}\left(\frac{\hat{y}_i - y_i}{y_i}\right) \times 100\% \tag{9}$$

Where n denotes the number of observations, $\hat{y}_i - y_i$ denotes the error between the forecasted and actual value. The mean of the actual forecasting error is calculated by taking the arithmetic average of the absolute errors between the prediction and the actual value. The root mean square error (RMSE) is a commonly used metric for comparing the values forecasted by a model or estimate to the values observed, and it's the average squared error squared. The MAPE measure calculates accuracy as a percentage, computed as the actual values minus the forecasted values divided by the actual values for each time period [46].

## Data analysis

Statistical analyses were performed using RStudio (Version 4.1.0) [47]. The 'tseries', 'TSstudio' and stats packages were used to process the time series. ARIMA models were built with the 'forecast' package using auto.arima function for choosing the best model based on the AICc values [48]. The 'forecastxgb' package was used for fitting XGBoost model. The necessary codes are available at https://github.com/ [49].

## Results

In Bangladesh, 1.6 million cases and 27,983 deaths of COVID-19 of 633 days (91 weeks) were recorded from March 08, 2020 to November 30, 2021. The highest COVID-19 confirmed cases were recorded 16,230 and deaths 264 in Bangladesh (Table 1). The data vary considerably and show weekly seasonality and nonlinearity pattern in both cases and deaths. Although the number of confirmed cases and deaths fluctuated in different weeks, there was a highly upward trend between 70 and 80 weeks. After that, it began to alleviate (Fig 2). The ADF test confirms that the data are not smooth. The entire data set (COVID-19 confirmed cases and deaths) was split into several training and test sets (S1 Text).

**Table 1. Summary of COVID-19 confirmed cases and deaths count during March 08- November 30, 2021.**

| Variables | Minimum | Maximum | Mean±SD | Total |
|---|---|---|---|---|
| Confirmed cases | 0 | 16,230 | 2490±2938.7 | 1,576,284 |
| Deaths | 0 | 264 | 45.2±54.5 | 27,981 |

SD: Standard deviation.

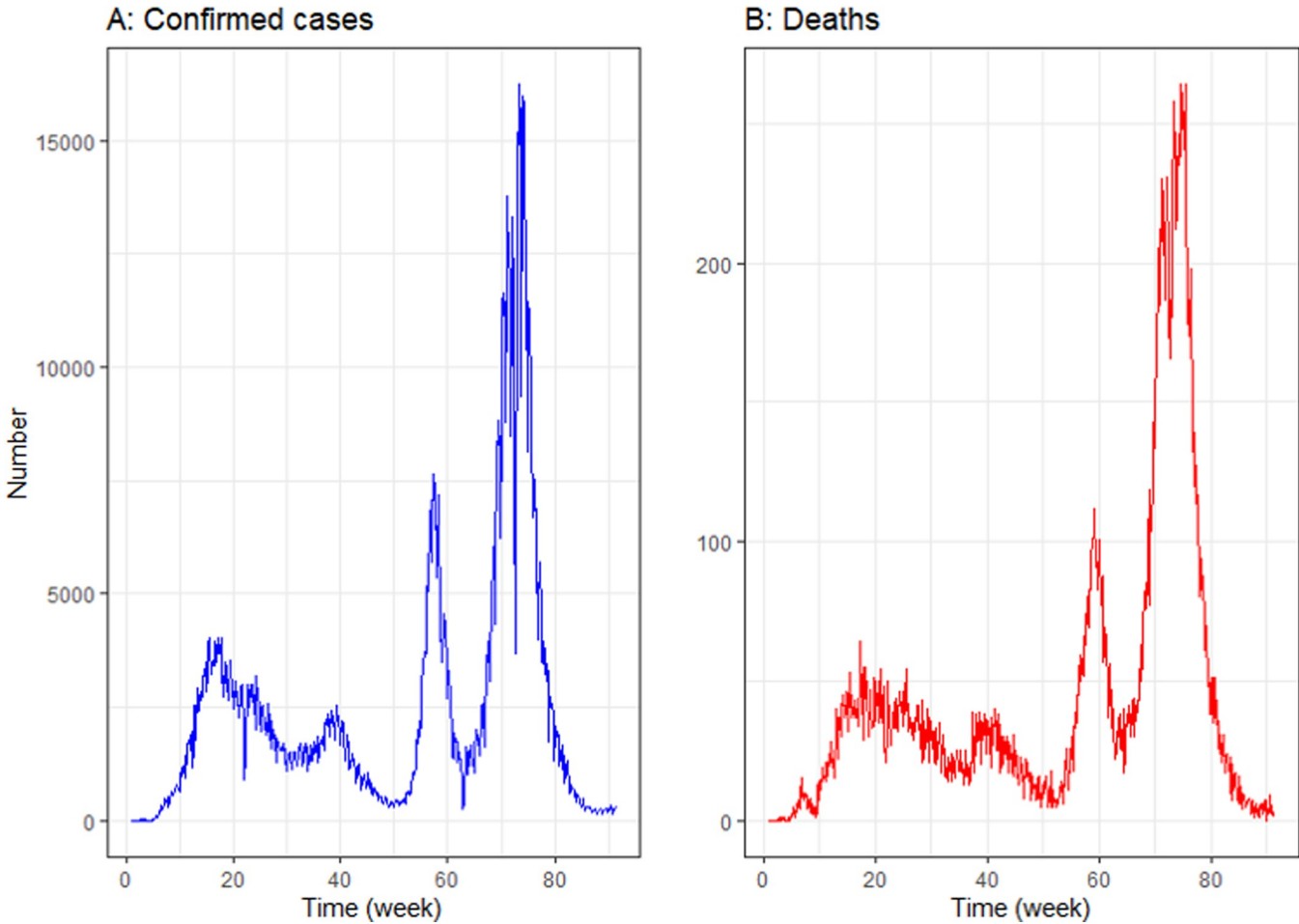

**Fig 2. A 633 day (91 weeks) time series plot for confirmed COVID-19 cases and deaths in Bangladesh from March 08, 2020 to November 30, 2021.**

To decrease anomalies such as non-normality and heteroscedasticity that the variances are not constant, Box & Cox (1964) presented a parametric power transformation technique [50]. The Box-Cox transformation was applied to each training data set to remove the non-normality and exhibit less variation [51]. The decomposed data shows a weekly seasonal pattern in both cases and deaths [52]. Table 2 illustrates the predictive performance of different ARIMA models built from seven different training sets and their average values for COVID-19 confirmed cases.

The XGBoost model for COVID-19 confirmed cases were built by adjusting different parameters like seas_method= 'dummies', trend_method= 'none', power transformation parameter 'lambda' for each training set. Table 3 illustrates the predictive performances of different training and test sets of the XGBoost model and their average values for confirmed cases.

For COVID-19 death data, we built five different ARIMA models for five different training and test sets. The appropriate model for each training data set was selected based on the AICc value. The predictive performances of ARIMA models of five different training and test data sets and their average values were shown in Table 4.

We also built the XGBoost model for COVID-19 deaths by adjusting the parameters seas_-method= 'dummies', trend_method= 'none', power transformation parameter 'lambda' for

**Table 2. Evaluation of parameters for the ARIMA model of different training and test sets for COVID-19 confirmed cases.**

| ARIMA model | Train | | | | Test | | | |
|---|---|---|---|---|---|---|---|---|
| | RMSE | MAE | MPE | MAPE | RMSE | MAE | MPE | MAPE |
| Sample 1 | 258.50 | 166.16 | -2.53 | 15.00 | 421.92 | 359.07 | -2.63 | 22.46 |
| Sample 2 | 241.82 | 154.73 | -2.72 | 13.39 | 244.33 | 216.94 | -47.26 | 48.85 |
| Sample 3 | 224.11 | 139.44 | -2.67 | 12.80 | 3844.89 | 2988.57 | 75.95 | 75.95 |
| Sample 4 | 280.81 | 165.97 | -1.72 | 12.50 | 2160.95 | 2031.37 | -163.43 | 163.44 |
| Sample 5 | 275.28 | 173.14 | -1.96 | 12.82 | 6325.54 | 5481.82 | 49.68 | 50.74 |
| Sample 6 | 560.19 | 276.18 | -2.15 | 13.11 | 8984.24 | 8217.15 | -549.35 | 549.35 |
| Sample 7 | 558.91 | 280.01 | -2.47 | 12.82 | 67.31 | 55.24 | 1.02 | 21.36 |
| Average error measures | 342.80 | 193.66 | -2.32 | 13.21 | 3149.88 | 2764.31 | -90.86 | 133.16 |

ARIMA: Autoregressive Integrated Moving Average; RMSE: Root Mean Square Error; MAE: Mean Absolute Error; MPE: Mean Percentage Error; MAPE: Mean Absolute Percentage Error.

each training set. The predictive measures of different training and test set for XGBoost model and their average values for COVID-19 deaths were shown in Table 5.

The average MAPE values of the ARIMA model for COVID-19 confirmed cases is comparatively lower than the XGBoost model indicating that ARIMA performs better than XGBoost in predicting COVID-19 confirmed cases in Bangladesh. On the other hand, the average MAPE value of the ARIMA model for COVID-19 deaths is smaller than XGBoost which also indicates that ARIMA performs better than XGBoost in predicting COVID-19 deaths in Bangladesh.

In our study, it was found that ARIMA model performs better than XGBoost in predicting COVID-19 confirmed cases and deaths in Bangladesh. The detailed procedure of ARIMA and XGBoost model fitting for COVID-19 confirmed cases and deaths were shown in S1 Text.

## Discussion

In our study, we found a weekly seasonality for daily COVID-19 confirmed cases and deaths in Bangladesh. Because of the weekend, fewer health care staffs were available to report new cases or fewer people are tested, which causes weekly seasonality [11, 53]. It was simpler to assess the seasonality and pattern of this disease using seasonal decomposition, which offered a reference for us to analyze, process, and stabilize data, laying the groundwork for building a

**Table 3. Evaluation of parameters for the XGBoost model of different training and test sets for COVID-19 confirmed cases.**

| XGBoost model | Train | | | | Test | | | |
|---|---|---|---|---|---|---|---|---|
| | RMSE | MAE | MPE | MAPE | RMSE | MAE | MPE | MAPE |
| Sample 1 | 47.19 | 31.66 | -0.11 | 2.30 | 520.76 | 436.20 | 11.44 | 23.97 |
| Sample 2 | 31.39 | 22.05 | -0.16 | 1.72 | 925.60 | 865.12 | -190.79 | 190.81 |
| Sample 3 | 64.91 | 42.74 | 0.36 | 3.10 | 3727.73 | 2874.31 | 71.05 | 71.12 |
| Sample 4 | 76.87 | 51.28 | -0.11 | 3.64 | 1989.87 | 1814.20 | -156.36 | 156.56 |
| Sample 5 | 53.71 | 35.71 | -0.24 | 2.53 | 7374.24 | 6413.23 | 59.46 | 59.46 |
| Sample 6 | 130.08 | 80.66 | -0.17 | 4.06 | 9683.20 | 9183.94 | -561.51 | 561.55 |
| Sample 7 | 168.82 | 105.89 | -0.14 | 4.64 | 196.60 | 185.76 | -76.20 | 76.20 |
| Average error measures | 81.85 | 52.86 | -0.08 | 3.14 | 3488.29 | 3110.39 | -120.42 | 162.81 |

XGBoost: eXtreme Gradient Boosting; RMSE: Root Mean Square Error; MAE: Mean Absolute Error; MPE: Mean Percentage Error; MAPE: Mean Absolute Percentage Error.

**Table 4. Evaluation of parameters for the ARIMA models of different training and test sets for COVID-19 deaths.**

| ARIMA model | Train | | | | Test | | | |
|---|---|---|---|---|---|---|---|---|
| | RMSE | MAE | MPE | MAPE | RMSE | MAE | MPE | MAPE |
| Sample 1 | 5.84 | 4.35 | -6.69 | 23.46 | 46.32 | 33.48 | 49.72 | 69.15 |
| Sample 2 | 6.14 | 4.59 | -4.71 | 23.09 | 110.31 | 101.42 | -287.79 | 286.79 |
| Sample 3 | 7.23 | 5.37 | -4.64 | 24.34 | 100.32 | 90.70 | 47.81 | 47.81 |
| Sample 4 | 9.98 | 6.67 | -4.35 | 22.51 | 246.09 | 228.36 | -620.60 | 620.60 |
| Sample 5 | 9.64 | 6.60 | -4.98 | 21.22 | 6.00 | 5.40 | -152.45 | 154.40 |
| Average error measures | 7.77 | 5.52 | -5.07 | 22.92 | 101.81 | 91.87 | -192.66 | 235.75 |

ARIMA: Autoregressive Integrated Moving Average; RMSE: Root Mean Square Error; MAE: Mean Absolute Error; MPE: Mean Percentage Error; MAPE: Mean Absolute Percentage Error.

mathematical model. The ARIMA models for cases and deaths were created using a linear regression model to expose the data's dynamic rules and forecast future data values. The ARIMA model combines the trend components, cyclical factors, and random errors originally included in the time series. This model combines the benefits of autoregressive and moving average models, is unconstrained by data sources, has high adaptability, and has good short-term predictions [18]. Instead of requiring particular influencing elements, the ARIMA model uses merely historical data to comprehend the illness pattern and achieve a more accurate forecast impact. As a result, the ARIMA approach is simple to learn and frequently employed [38]. In this study, the ARIMA method is compared to the XGBoost model for its fairly mature time series prediction approach and widespread application. The ARIMA model performs well on the nonstationary time series after applying Box-Cox transformation and differencing adjustments, demonstrating the model's capacity to forecast diseases. In general, the greater the number of differences utilized, the more data is lost. We built different ARIMA models for different training sets and selected the best for each training set based on the AICc value for both confirmed cases and deaths [53, 54]. Finally, we averaged all the error measures from all models. The average MAPE value of the training data sets for confirmed cases was 13.21%, whereas it was 133.16% for the test data sets. On the other hand, the average MAPE value of the training sets for death data was 22.92%, whereas the test sets was 235.95%. On the other hand, we used the most popular machine learning model to fit the nonlinear data [55]. The XGBoost model, a relatively new approach, is a gradient boosting-based ensemble machine learning technique that utilizes decision trees. The XGBoost technique offers several benefits in terms of model prediction, including the lack of data preprocessing, a quick operation speed, complete feature

**Table 5. Evaluation of parameters for the XGBoost models of different training and test sets for COVID-19 deaths.**

| XGBoost model | Train | | | | Test | | | |
|---|---|---|---|---|---|---|---|---|
| | RMSE | MAE | MPE | MAPE | RMSE | MAE | MPE | MAPE |
| Sample 1 | 2.19 | 1.45 | 0.32 | 6.34 | 40.32 | 28.11 | 20.68 | 63.18 |
| Sample 2 | 1.95 | 1.39 | -1.03 | 6.65 | 49.17 | 45.01 | -131.30 | 132.18 |
| Sample 3 | 3.80 | 2.66 | 1.69 | 10.05 | 150.98 | 136.59 | 71.82 | 72.82 |
| Sample 4 | 2.70 | 1.92 | -1.49 | 7.37 | 179.88 | 169.85 | -444.68 | 445.68 |
| Sample 5 | 3.20 | 2.27 | -1.18 | 7.50 | 16.27 | 15.47 | -470.79 | 471.27 |
| Average error measures | 2.77 | 1.94 | -0.34 | 7.58 | 87.32 | 79.01 | -190.85 | 237.03 |

XGBoost: eXtreme Gradient Boosting; RMSE: Root Mean Square Error; MAE: Mean Absolute Error; MPE: Mean Percentage Error; MAPE: Mean Absolute Percentage Error.

extraction, a strong fitting effect, and high prediction accuracy. This study applied this new technique to predict COVID-19 confirmed cases and deaths in Bangladesh. We selected the most often used ARIMA time series model as the baseline of this study. But the XGBoost model did not perform well on the nonlinear data. The XGBoost model has a considerably worse influence on forecasting than the ARIMA model in this COVID-19 research in Bangladesh because the number of confirmed cases and deaths has increased significantly between 70 and 80 weeks. The number of confirmed cases in the country has also altered dramatically due to changes in government policies. In addition, there might have other climatic and environmental factors that impact the COVID-19 incidence observed from some previous studies which didn't incorporate in our study [46, 56–58]. As a result, the proposed model was no longer produced accurate predictions for this change. In this study, we compared the models' predictive performances to provide a reference for the country's policymakers to take effective steps and strategies to control the outbreak of the deadly disease. The study findings are useful to all other endemic countries similar to Bangladesh.

## Conclusion

For controlling the spread of the COVID-19 pandemic in Bangladesh and similar settings elsewhere, we developed a seasonal ARIMA model and XGBoost model. These models were used to create short-term forecasts in this study. The ARIMA model performed better than the XGBoost model in predicting COVID-19 confirmed cases and deaths in Bangladesh.

## Limitations

We compared the predictive performance of XGBoost and ARIMA models in this study, and the results help choose the best model for COVID-19 prediction in Bangladesh. There are many different prediction models, and we need to keep experimenting with them to find the best one for predicting confirmed COVID-19 cases and deaths. We focused on the impact of time on both cases and deaths in our research, which allows our model easier to build and forecast. Therefore, a limitation of our study is that, for example, meteorological data such as temperature, humidity, and wind speed variables were not incorporated but which are known to impact COVID-19. As mentioned above, this will be explored progressively with increasing data.

## Supporting information

**S1 Data. Time series COVID-19 data of Bangladesh from March 08, 2020 to November 30, 2021.**
(XLSX)

**S1 Text. Figs A–J, Tables A–B.**
(DOCX)

## Acknowledgments

The researchers are very grateful to the Directorate General of Health Service (DGHS) and the Institute of Epidemiology, Disease Control and Research (IEDCR) for providing COVID-19 data.

## Author Contributions

**Conceptualization:** Md. Siddikur Rahman.

**Data curation:** Md. Siddikur Rahman, Arman Hossain Chowdhury, Miftahuzzannat Amrin.

**Formal analysis:** Md. Siddikur Rahman, Arman Hossain Chowdhury.

**Investigation:** Md. Siddikur Rahman, Arman Hossain Chowdhury.

**Methodology:** Md. Siddikur Rahman, Arman Hossain Chowdhury.

**Project administration:** Md. Siddikur Rahman.

**Resources:** Md. Siddikur Rahman, Arman Hossain Chowdhury, Miftahuzzannat Amrin.

**Software:** Md. Siddikur Rahman, Arman Hossain Chowdhury.

**Supervision:** Md. Siddikur Rahman.

**Visualization:** Md. Siddikur Rahman, Arman Hossain Chowdhury.

**Writing – original draft:** Md. Siddikur Rahman, Arman Hossain Chowdhury.

**Writing – review & editing:** Md. Siddikur Rahman, Arman Hossain Chowdhury, Miftahuzzannat Amrin.

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
