## [Decision Letter · Decision Letter 0]

5 Nov 2021

PGPH-D-21-00586

Accuracy comparison of ARIMA and XGBoost forecasting models in predicting the incidence of COVID-19 in Bangladesh

Dear Dr. Rahman,

Thank you for submitting your manuscript to PLOS Global Public Health. After careful consideration, we feel that it has merit but does not fully meet PLOS Global Public Health’s publication criteria as it currently stands. Therefore, we invite you to submit a revised version of the manuscript that addresses the points raised during the review process.

Thank you very much for this submission. As you will see there are some substantial requests from the reviewers about revisions that I think will make this paper stronger and more impactful. I would like to emphasize the importance of one line of reviewer suggestions, about making the data from your analysis easily available---I recommend you go one step further and make a "replication archive" available. Such an archive would include the data as well as the analysis code necessary to reproduce the key results in your paper. This practice can help accelerate science and drive further dissemination of your research.

I hope you will find the reviewer reports constructive and that they will find a revised version of your paper to be a valuable contribution to the scientific literature. Thank you again for your work on this important topic.

We look forward to receiving your revised manuscript.

Kind regards,

Abraham D. Flaxman, Ph.D.

Academic Editor

Journal Requirements:

1. Please provide separate figure files in .tif or .eps format only, and remove any figures embedded in your manuscript file.  If you are using LaTeX, you do not need to remove embedded figures.

2. Please update the completed 'Competing Interests' statement, including any COIs declared by your co-authors. If you have no competing interests to declare, please state "The authors have declared that no competing interests exist". Otherwise please declare all competing interests beginning with the statement "I have read the journal's policy and the authors of this manuscript have the following competing interests:

3. Please provide a complete Data Availability Statement in the submission form, ensuring you include all necessary access information or a reason for why you are unable to make your data freely accessible. Note that it is not acceptable for the authors to be the sole named individuals responsible for ensuring data access.

PLOS defines a study's minimal data set as the underlying data used to reach the conclusions drawn in the manuscript and any additional data required to replicate the reported study findings in their entirety. Any potentially identifying patient information must be fully anonymized. 

If your research concerns only data provided within your submission, please write ""All data are in the manuscript and/or supporting information files"" as your Data Availability Statement.

4. Please provide a detailed Financial Disclosure statement. This is published with the article, therefore should be completed in full sentences and contain the exact wording you wish to be published.

i) Please include all sources of funding (financial or material support) for your study. List the grants (with grant number) or organizations (with url) that supported your study, including funding received from your institution. 

ii). State the initials, alongside each funding source, of each author to receive each grant.

iii). State what role the funders took in the study. If the funders had no role in your study, please state: “The funders had no role in study design, data collection and analysis, decision to publish, or preparation of the manuscript.”

iv). If any authors received a salary from any of your funders, please state which authors and which funders.

Additional Editor Comments (if provided):

Thank you very much for this submission. As you will see there are some substantial requests from the reviewers about revisions that I think will make this paper stronger and more impactful. I would like to emphasize the importance of one line of reviewer suggestions, about making the data from your analysis easily available---I recommend you go one step further and make a "replication archive" available. Such an archive would include the data as well as the analysis code necessary to reproduce the key results in your paper. This practice can help accelerate science and drive further dissemination of your research.

I hope you will find the reviewer reports constructive and that they will find a revised version of your paper to be a valuable contribution to the scientific literature. Thank you again for your work on this important topic.

Reviewers' comments:

Reviewer's Responses to Questions

**Comments to the Author**

1. Does this manuscript meet PLOS Global Public Health’s publication criteria? Is the manuscript technically sound, and do the data support the conclusions? The manuscript must describe methodologically and ethically rigorous research with conclusions that are appropriately drawn based on the data presented.

Reviewer #1: Partly

Reviewer #2: No

2. Has the statistical analysis been performed appropriately and rigorously?

Reviewer #1: I don't know

Reviewer #2: No

3. Have the authors made all data underlying the findings in their manuscript fully available (please refer to the Data Availability Statement at the start of the manuscript PDF file)?

Reviewer #1: No

Reviewer #2: Yes

4. Is the manuscript presented in an intelligible fashion and written in standard English?

Reviewer #1: Yes

Reviewer #2: No

5. Review Comments to the Author

Reviewer #1: Review of “Accuracy comparison of ARIMA and XGBoost forecasting models in predicting the incidence of COVID-19 in Bangladesh”

Thank you for the opportunity to review this manuscript. I found it interesting and enjoyed reading it. I think work assessing the predictive performance of COVID-19 forecasting models is needed, especially as we are now starting to reflect on what has gone well/poorly in this pandemic management. I do have some concerns about the overall vision for this analysis, as well as a few key analytical decisions. I think this work is not publishable in its current form, but has potential if the authors wish to undertake a considerable amount of work to provide new analyses and revisions to the framing of the text.

Major Comments:

1. My biggest concern is that it seems problematic to me that a single holdout period was chosen. It has been shown that predictive performance massive varies by time period, even for the same model using the same methods. Models do not perform equally during the various sets of circumstances that can arise (trends decreasing/increasing, speed of change, overall shape, etc.). I would recommend running this analysis on a set of training/test sets, iterating over all possible cutoff dates, and presenting aggregated predictive performance results, perhaps also stratified by month or season.

2. It is also not clear to me why a 1 week testing set was used. Longer-range forecasts are likely much more useful for policymakers. I would recommend seeing how these models perform at least 4 weeks out, but arguably even much longer range forecasts, like 8 to 12 seeks, are much more actionable from a policy perspective.

3. Why were cases chosen and not deaths as the outcome measure of interest? Both are flawed measures depending on health system and surveillance factors, but deaths are generally much more reliable. Cases can swing wildly based on shifts in testing rates, etc.

4. I would recommend also assessing median error or median percent error, a measure of bias, alongside measures of accuracy.

5. Overall, I felt there was a lack of interpretation of the implications of the results for the real world. One model was found to have a better predictive performance. But more perspective is needed on why that is consequential. Thinking through why this work matters, and for whom, can also help sharpen the analytical choices.

6. For example, why where these models compared only on data from Bangladesh. If this paper is about these methods, then comparing them on data from many more countries would provide a much more robust analysis (and it does not take much more work to compare 1 country or 100, given that these analyses are conducted in R). But if this paper is about the unique context of Bangladesh, and providing information for policymakers from that country, then that needs to be clearer.

7. I think there is too much emphasis on the in-sample predictive performance of these models, which isn’t necessarily important for a forecasting model.

8. Data and code availability is not specified in the piece (that I can see) which would be a violation of journal requirements for publication.

Minor Comments:

1. In the figures, the time scale makes it difficult to see the actual predictions vs. subsequently observed reality. I would recommend showing fewer data points to make the predictions more legible. Especially since the im-sample fit doesn’t really matter for the main goals of this paper, unless I am misunderstanding them.

2. Some of the details about how models were fit (e.g. auto.arima() used in R) were not available in methods, where I expected them, but rather the results section, which made me initially think they were missing

Reviewer #2: This study looks at the COVID-19 epidemic in Bangladesh. COVID-19 cases from March 2020 to August 2021 were retrieved and divided into a training [March 2020-August 19 2021] and test [August 20 2021, August 26 2021] sets. The authors fit the data with an autoregressive integrated moving average (ARIMA) and a regularised gradient boosting (XGBoost) models on the training set and compare their prediction performances on the test set. Authors conclude that the ARIMA is not flexible enough to fit well the data while the XGBoost obtains encouraging prediction performance.

MAJOR COMMENTS

- My biggest concern about this paper as it currently stands is the lack of a clear link making it of direct relevance to public health. While the authors explain the catastrophic consequences of COVID-19 on Bangladesh’s economy, education and health care, they do not demonstrate how predictions from their models regarding confirmed cases could have been used to tackle the crisis or provide useful actionable information to policymakers. Further explanation and discussion (both of this and the limitations of their approach) is therefore required.

- A second issue is that confirmed cases are not easily interpretable - they depend on many time-varying factors (such as testing availability, healthcare seeking behaviours etc). It is unclear then whether fitting these models to case data is picking up legitimate epidemiological signals. It is this reason that has motivated many epidemiological models to use COVID-19 deaths as an indicator of the progression and trajectory of SARS-CoV-2 epidemics (see e.g. Flaxman et al., Nature). The authors might consider fitting their models to deaths instead of case data, though note (see below) that irrespective of the underlying dataset used, I also have significant concerns about the statistical approach adopted.

- The time-period spanned by the test dataset (August 20th to 26th 2021) is very small compared to the period spanned by the training dataset (March 1st 2020 to August 19th 2021). Epidemiological quantities (such as the reproduction number) driving the dynamics of the case counts are unlikely to fluctuate substantially over such short time-periods – it is therefore unclear whether the models are successfully capturing the underlying epidemiological phenomena driving temporal fluctuations in case counts. It is necessary to explore a range of training and test data set sizes, and particularly, for the authors to evaluate model performance over a time period where the reproduction number clearly changes (e.g. weeks 45-55 or 60-70 for example where the declining case curve then goes back up again).

- There is currently a lack of explanation and description of the chosen statistical methods (and the inference approach i.e. the algorithms used to fit the model). Whilst models similar to XGBoost have been used for time-series forecasting, this is typically done by first generating/collating various features potentially relevant to the dataset (such as day-of-the-week, seasonality etc) and then training the model using these features. It is not clear currently what the authors did as the statistical explanations are too brief and not specific enough. Moreover, if the authors did not use this feature-based approach and instead trained it directly on the time-series data itself, I have significant concerns as to the validity of that choice (I am not aware of other published examples of approaches like this and am unsure whether such an approach is valid).

- The authors claim that XGBoost has good prediction accuracy – description of this is quite limited currently however, and only indirectly quantified through comparison to ARIMA.

DETAILED COMMENTS

ARIMA:

- Equation (1): why the ‘y’ on the LHS is in capital and not the ‘y’s on the RHS?

- I do not follow how the model is defined. The authors state that the model is decomposed in two parts, a ‘simple seasonal model’ (in eq 4) and a ‘product seasonal model’. Where is eq.3 used? And how are S_t, T_t and I_t found?

Gradient boosting model:

- Gradient boosting has been shown to be efficient in cross-sectional data. The authors lack to reference/demonstrate why using this method on time series makes sense theoretically.

- “The model may need to be iterated multiple times or more”: the authors must define what iterated means in this context.

- ‘cost function’ à ‘loss function’

- Equation (5): What is x_i? Similarly in this equation and all that apply, I understand what y_i means from context but I do not think it is defined anywhere.

- Equation (5): It is hard to follow this equation, f_m has input x_i in one instance and not the next, and what is the operation underlined in (y_i, y_i^{m-1})? Lastly the loss function has one input and not two? Do the authors meant L[y_i, (y_i^{m-1} + f_m(x_i))]?

- What are the covariates (or features) used to fit this model? The week indices?

- The authors do not provide sufficient detail about the way in which parameters of the gradient boosting model were optimised – if fixed parameters were eventually chosen, that is fine, but it should be clearly stated. If hyperparameter optimisation was undertaken, then an explanation about how this was carried out is required.

Evaluation parameter of models:

- This section states elementary statistics and could be moved to the supplements or removed.

- Why do the authors desire to compare three different metrics? Why not pick only one?

Application ARIMA:

- The motivation for the Box-Cox transform is not clear “to make the dataset more stable”. What is the definition of stable in this context? Especially because The Box-Cox transform creates two huge drops at the beginning of the sequence. And the large increase at the end of the sequence is now almost at the same magnitude as the first wave. Could the authors provide a sensitivity analysis where the original data are fitted? Are the original data just tweaked to fit the assumptions of the ARIMA?

- What is the cause of the seasonal effects according to the authors? They briefly discuss environmental effects in the discussion without more explanation on their relation to confirmed cases. Could it be that there is a drop in the week-ends because less health care workers are employed to report the new cases/less people get tested, and the spike at the beginning of the week to account for reporting delays? (underlying again the issues arising from using confirmed cases data)

- What is the difference between the AIC and the AICc? The authors mention both

- Why the AIC is used for model selection of the ARIMA and other metrics for model comparison between the ARIMA and gradient boosting?

- ‘The arima model was built with the auto.’ Is this sentence not finished somehow?

Application Gradient boosting:

- ‘The XGBoost model with the greatest results was developed by changing the parameters frequently’, the authors should clarify what this means in context, do they fit the model weekly or monthly?

Prediction:

The prediction part is not convincing. The authors fit the entire data set up to the before to last week and only predict the confirmed number of cases in the last week.

- Figure 7 and 8 do not convince me of the efficacy of any of their methods for prediction as their point estimates are off.

- Even more concerning is the prediction by the ARIMA of a negative number of cases at the beginning of the sequence.

- Do the authors have an explanation as to why the XGBoost method does not manage to predict well the spikes included the training set towards the end of the data set?

- Could the authors show the confidence intervals of their predictions?

- Could the authors show multiple prediction scenarios where more days are predicted?

- Most importantly theoretically, the authors lack to demonstrate how the information collected in the first weeks inform the predictions at week 70+? (1) Up to a certain point, the information of the past has no impact on the future, and therefore most of the dataset could be excluded and (2) what epidemiological signal is learned.

Discussion:

- The authors argue for a beneficial effect of the ‘autoregressive and average moving models’. A more thorough discussion of those benefits is necessary.

- “the training and test sets performed very differently”, a data set does not perform.

- The evidence for calling gradient boosting’s performance ‘outstanding’ are not strong. In order to make this claim, the predictions obtained with gradient boosting should be compared to more methods that could in theory perform well (i.e., not ARIMA) and previous studies that tackled the same problem.

Figures

- Informative axes title (e.g., ‘Confirmed COVID-19 attributable cases’ instead of ‘cases’ for Figure 1) would be appreciated

- Dating the y axis would be more informative than the week indexing.

- Figure 6 does not have a y-axis title on the top plot

- Labelling plots by A, B, C in Figure 6 would make the explanation of the figure clearer.

- Self explanatory caption and columns label for table 1 would be useful

Typos:

- I spotted some ‘covid-19’ instead of ‘COVID-19’

- Some ‘arima’ instead of ‘ARIMA’

- Some sentences would benefit polishing, e.g., ‘the number of cases is increasing significantly after 400 days from the beginning in the country’.

- Some terms should be chosen more rigorously, e.g., ‘fitting and prediction impact’

References

Flaxman, S., Mishra, S., Gandy, A. et al. Estimating the effects of non-pharmaceutical interventions on COVID-19 in Europe. Nature 584, 257–261 (2020). https://doi.org/10.1038/s41586-020-2405-7

6. PLOS authors have the option to publish the peer review history of their article (what does this mean?). If published, this will include your full peer review and any attached files.

**Do you want your identity to be public for this peer review?** For information about this choice, including consent withdrawal, please see our Privacy Policy.

Reviewer #1: No

Reviewer #2: No

---

## [Decision Letter · Decision Letter 1]

14 Jan 2022

PGPH-D-21-00586R1

Accuracy comparison of ARIMA and XGBoost forecasting models in predicting the incidence of COVID-19 in Bangladesh

Dear Dr. Rahman,

Thank you for submitting your manuscript to PLOS Global Public Health. After careful consideration, we feel that it has merit but does not fully meet PLOS Global Public Health’s publication criteria as it currently stands. Therefore, we invite you to submit a revised version of the manuscript that addresses the points raised during the review process.

We look forward to receiving your revised manuscript.

Kind regards,

Abraham D. Flaxman, Ph.D.

Academic Editor

Journal Requirements:

Additional Editor Comments (if provided):

Thank you for this extensive review. As you will see you moved your most negative reviewer from "Reject" to "Major Revision" which is an important step forwards. Please continue to take all of the feedback from these reviewers very seriously.

Reviewers' comments:

Reviewer's Responses to Questions

**Comments to the Author**

1. If the authors have adequately addressed your comments raised in a previous round of review and you feel that this manuscript is now acceptable for publication, you may indicate that here to bypass the “Comments to the Author” section, enter your conflict of interest statement in the “Confidential to Editor” section, and submit your "Accept" recommendation.

Reviewer #1: All comments have been addressed

Reviewer #2: (No Response)

2. Does this manuscript meet PLOS Global Public Health’s publication criteria? Is the manuscript technically sound, and do the data support the conclusions? The manuscript must describe methodologically and ethically rigorous research with conclusions that are appropriately drawn based on the data presented.

Reviewer #1: Partly

Reviewer #2: Yes

3. Has the statistical analysis been performed appropriately and rigorously?

Reviewer #1: Yes

Reviewer #2: No

4. Have the authors made all data underlying the findings in their manuscript fully available (please refer to the Data Availability Statement at the start of the manuscript PDF file)?

Reviewer #1: Yes

Reviewer #2: Yes

5. Is the manuscript presented in an intelligible fashion and written in standard English?

Reviewer #1: Yes

Reviewer #2: Yes

6. Review Comments to the Author

Reviewer #1: The authors have undertaken considerable work to update the analyses, which are greatly appreciated, and have strengthened the research. However, the work still uses a single training and a single testing period. Looking at Figures 7 and 8 it become clear why this is not appropriate. The forecasts are assessed relative to only a tiny window of the overall pandemic, and at a time when rates where monotonically decreasing. These results will likely not be generalizable to the remainder of the timeseries where deaths are rising or peaking. Instead, for this work to be robust, the authors must average predictive performance over a series of testing and training holdouts. This would entail iterating over all possible start dates (e.g. each week from May 2020 to October 2021), fitting the model on all available data, and checking predictive performance on the remainder of the timeseries. Predictive validity statistics can then be averages across all the various testing windows.

All other edits made by the authors are satisfactory.

Reviewer #2: The manuscript has been improved most notably by the inclusion of the predictions of confirmed covid-19 deaths. I also remarked an improvement of the writing and the motivations for predicting covid-19 deaths. And I commend the authors for both. That being said, the paper still falls short, in my opinion, in a clear explanation of the methods and the algorithms. The maths of the methods is not clear or incorrect, and the algorithm used for fitting is not stated. Saying that one used a specific package is not enough in this context. Some comments were only partly addressed or entirely ignored.

The conclusions of the authors did change between the previous version and the revision. I believe due to the extension of the test period. ARIMA is now found to perform better. The fact that the gradient boosting method does not perform better is not problematic for me. Even bad results are important for science to progress. However, gradient boosting must be fed with information (in the same way that the ARIMA has a specific parametric form which induces seasonal trends). And it is still not clear to me what information (i.e., features) have been given to the gradient boosting. In this context I cannot be convinced of the poor performance of gradient boosting.

ARIMA MODEL

-Are X_t covariates of the seasonal model?

-To my best knowledge, the authors did not explain what P and Q were in l.153-154 and the authors did not state what d from ARIMA(p, d, q) governed and how it was estimated.

-In the response to reviewers, the authors did not specify rigorously how to find S_t, T_t and I_t

-Eq (2) and (3) it should be theta_q not theta_p.

XBOOST

- A Loss function in (5)is defined by two outputs and may be written as L(y, ybar), which is still not the case.

“- What are the covariates (or features) used to fit this model? The week indices?”. Their answer “no covariates were present here” However in the text they say “x_i is the feature vector”. So again, what are the features?

7. PLOS authors have the option to publish the peer review history of their article (what does this mean?). If published, this will include your full peer review and any attached files.

**Do you want your identity to be public for this peer review?** For information about this choice, including consent withdrawal, please see our Privacy Policy.

Reviewer #1: No

Reviewer #2: No

---

## [Decision Letter · Decision Letter 2]

28 Feb 2022

PGPH-D-21-00586R2

Accuracy comparison of ARIMA and XGBoost forecasting models in predicting the incidence of COVID-19 in Bangladesh

Dear Dr. Rahman,

Thank you for submitting your manuscript to PLOS Global Public Health. After careful consideration, we feel that it has merit but does not fully meet PLOS Global Public Health’s publication criteria as it currently stands. Therefore, we invite you to submit a revised version of the manuscript that addresses the points raised during the review process.

As you will see, the reviewers still have major concerns with your approach, and I feel that I cannot keep sending them minor improvements from you.  In light of this, I will need you to ensure that you include out-of-sample predictive performance averaged across a wide range of training/testing splits in the main text, as requested by Reviewer 1.

I also find Reviewer 2's concerns convincing and I will need you to clearly frame your work as a comparison specifically of ARIMA and a limited XGBoost where the only covariate is the week index (or, if you are able, follow Reviewer 2's suggestion and give additional information to the XGBoost model by using as covariate features potentially relevant to the dataset (such as day-of-the-week and seasonality).

We look forward to receiving your revised manuscript.

Kind regards,

Abraham D. Flaxman, Ph.D.

Academic Editor

Journal Requirements:

Additional Editor Comments (if provided):

Reviewers' comments:

Reviewer's Responses to Questions

**Comments to the Author**

1. If the authors have adequately addressed your comments raised in a previous round of review and you feel that this manuscript is now acceptable for publication, you may indicate that here to bypass the “Comments to the Author” section, enter your conflict of interest statement in the “Confidential to Editor” section, and submit your "Accept" recommendation.

Reviewer #1: (No Response)

Reviewer #2: (No Response)

2. Does this manuscript meet PLOS Global Public Health’s publication criteria? Is the manuscript technically sound, and do the data support the conclusions? The manuscript must describe methodologically and ethically rigorous research with conclusions that are appropriately drawn based on the data presented.

Reviewer #1: Partly

Reviewer #2: Partly

3. Has the statistical analysis been performed appropriately and rigorously?

Reviewer #1: I don't know

Reviewer #2: N/A

4. Have the authors made all data underlying the findings in their manuscript fully available (please refer to the Data Availability Statement at the start of the manuscript PDF file)?

Reviewer #1: Yes

Reviewer #2: Yes

5. Is the manuscript presented in an intelligible fashion and written in standard English?

Reviewer #1: Yes

Reviewer #2: Yes

6. Review Comments to the Author

Reviewer #1: I appreciate the author's efforts to revise the piece. They have now provided a supplemental analysis using several different training and testing sets. However they have not changed the results in the main text which still rely on a single testing/training split. I would argue that the results shown in the main text (which were not revised on the last round, according to the tracked changes document) need to show the average across the various training/testing splits. Taking a look at their R code it appears to only run a single train/test split at a time. The reasons why this is insufficient were explained in my last round of review and are clear in the literature. Unless the authors can update the main text results to reflect the out-of-sample predictive performance averaged across a wide range of training/testing splits, I cannot support this work being published, because the results will simply be representative of one minor moment in a much larger pandemic.

Reviewer #2: The paper has been improved with the addition of new prediction analyses. However, in my opinion, the paper still requires much attention specifically in the methods section.

- The authors have made minor and insufficient changes regarding my previous comments on the need for clearer explanation on the different parts of the ARIMA model.

- Equation (3): shouldn’t there be only one epsilon_t?

- The authors STILL have not fixed the loss function. FYI reference [42] has a loss function which is formulated as l(x,y) (eq.6 in ref [42]).

- The authors stated that the only covariate of the GBoost were the week indices. It is therefore clear why GBoost doesn’t perform. On the one hand you have a heavily parametrised ARIMA model, that harnesses information from seasonal and autoregressive trends, and on the other hand you have a powerful GBoost that is fed with no information. How would the GBoost has any predictive performance using only week indices?

- The aim stated by this paper is already problematic. Because no model can make accurate prediction without incorporating epidemiological signals. So, the real aim of this paper is to compare, under partial information (i.e., here only the seasonal trends), which of the ARIMA or GBoost is capable of making the ‘best’ prediction (relative to the two models, not relative to models incorporating epidemiological trends proposed in the literature).

- In this context, (1) in my opinion the authors should rephrase the aim in the abstract and introduction to clearly express the above point. And (2) in order to make a fair comparison, it is essential that the authors find a way to give the same level of information to the two models. This could be achieved by using as covariate features potentially relevant to the dataset (such as day-of-the-week, seasonality etc). They can refer to their own citation such as [46].

I would like to finish my review by saying that all points above have been raised since the first round of reviews.

7. PLOS authors have the option to publish the peer review history of their article (what does this mean?). If published, this will include your full peer review and any attached files.

**Do you want your identity to be public for this peer review?** For information about this choice, including consent withdrawal, please see our Privacy Policy.

Reviewer #1: No

Reviewer #2: No

---

## [Editor Report · Decision Letter 3]

27 Apr 2022

Accuracy comparison of ARIMA and XGBoost forecasting models in predicting the incidence of COVID-19 in Bangladesh

PGPH-D-21-00586R3

Dear Mr. Rahman,

We are pleased to inform you that your manuscript 'Accuracy comparison of ARIMA and XGBoost forecasting models in predicting the incidence of COVID-19 in Bangladesh' has been provisionally accepted for publication in PLOS Global Public Health.

Best regards,

Abraham D. Flaxman, Ph.D.

Academic Editor